

# Increased game frequency period crossing Ramadan intermittent fasting decreases fat mass, sleep duration, and recovery in male professional basketball players

Seifeddine Brini[1,*], Luca Paolo Ardigò[2,*], Filipe Manuel Clemente[3,4,5], Javier Raya-González[6], Jennifer A. Kurtz[7], Gretchen A. Casazza[8], Carlo Castagna[9,10], Anissa Bouassida[1] and Hadi Nobari[11,12,13]

[1] Research Unit, Sportive Performance and Physical Rehabilitation, University of Jendouba, Kef, Tunisia
[2] Department of Teacher Education, NLA University College, Oslo, Norway
[3] Escola Superior Desporto e Lazer, Instituto Politécnico de Viana do Castelo, Viana do Castelo, Portugal
[4] Research Center in Sports Performance, Recreation, Innovation and Technology (SPRINT), Melgaço, Portugal
[5] Delegação da Covilhã, Instituto de Telecomunicações, Lisboa, Portugal
[6] Facultyof Sport Sciences, University of Extremadura, Cáceres, Spain
[7] Department of Kinesiology and Health, Georgia State University, Atlanta, GA, USA
[8] Department of Kinesiology, California State University, Sacramento, CA, USA
[9] Department of Biomolecular Sciences, School of Sport and Heath Sciences, University of Urbino "Carlo Bo", Urbino, Italy
[10] Department of Sports Science and Clinical Biomechanics, SDU Sport and Health Sciences Cluster (SHSC), University of Southern Denmark, Odense, Denmark
[11] Faculty of Sport Sciences, University of Extremadura, Caceres, Spain
[12] Department of Exercise Physiology, Faculty of Educational Sciences and Psychology, University of Mohaghegh Ardabili, Ardabil, Iran
[13] Department of Motor Performance, Faculty of Physical Education and Mountain Sports, Transilvania University of Brașov, Brașov, Romania
* These authors contributed equally to this work.

Corresponding authors
Seifeddine Brini,
bseifeddine15@gmail.com
Luca Paolo Ardigò,
luca.ardigo@nla.no

## ABSTRACT

**Background.** Increased basketball game frequency may affect athlete performances, especially during Ramadan intermittent fasting (RIF). The objective of the present investigation was to assess the impacts of increased game frequency periods crossing the RIF on body composition, sleep habits, indices of well-being, recovery state, and dietary intake in professional male basketball players.

**Methods.** Twenty-eight professional basketball players participated in this study and were divided into increased-games-frequency (INCR) or normal-games-frequency (NORM) groups. INCR trained four times and completed two games per week, whereas NORM completed only one game per week. During the first and fourth weeks of RIF, the following variables were assessed: internal load (weekly session rating of perceived exertion (s-RPE), heartrate (HR)), dietary intake, body composition, sleep quality (PSQI survey), well-being indices questionnaire (sleep, fatigue, stress, delayed onset of muscle soreness (DOMS)), and recovery state with the Total Quality Recovery (TQR) questionnaire.

**Results.** The internal load significantly increased after 4 weeks of RIF in INCR compared to NORM ($p < 0.001$). Significant decrease of TQR, sleep duration, and a significant

increase of DOMS only for INCR (26.93%, $p < 0.001$, ES = 0.48, small; 33.83%, $p < 0.001$, ES = 0.40, small; 161.17%, $p < 0.001$, ES = 0.32, small; respectively). Significant group × time interaction was observed for body mass ($p = 0.006$, ES = 0.46, small) and body fat percentage ($p = 0.025$, ES = 0.33, small), with INCR having a greater decrease in all these values.

**Conclusion**. Increased game frequency period crossing RIF decreases fat mass, sleep duration, and recovery in professional basketball players, which may consequently affect performance and health.

# INTRODUCTION

Basketball is an intermittently demanding team sport in which games are played in four quarters of 10 min (FIBA). These quarters are characterized by high-intensity movements interspersed with low-intensity activities (*Stojanović et al., 2018*). Additionally, the on-court success of this team sport depends on technical-tactical skill as well as physiological variables such as cardiovascular fitness, speed, agility, power, and strength (*Brien, Browne & Earls, 2020*; *Petway et al., 2020*; *Stojanović et al., 2018*). Previous studies have reported that performance measures (*e.g.*, shooting accuracy, decision-making and sports-specific motor performance (*Ardigò et al., 2018*; *Padulo et al., 2015*; *Padulo et al., 2018*)) can be negatively affected by an athlete's physiological or mental fatigue state (*e.g.*, overtraining, increased game frequency periods), or inadequate nutrition (*Brini et al., 2020*; *Clemente et al., 2019*; *Edwards et al., 2018*; *Fox et al., 2022*; *García et al., 2022*). In this context, several studies have monitored and quantified the physiological load (session rate of perceived exertion (s-RPE), heart rate (HR), blood lactate concentration, etc.) in basketball during training sessions and matches to manage the players' fatigue during any phase of the sports season (*Fox et al., 2022*; *Conte et al., 2018*; *Fox, Scanlan & Stanton, 2017*; *Moreno-Villanueva, Rico-González & Pino-Ortega, 2022*). Different circumstances could influence the s-RPE values in basketball, such as congested schedules with multiple matches played in close succession (*Edwards et al., 2018*), the weekly workloads encountered by players (*Conte et al., 2018*), and tapering strategies (*Edwards et al., 2018*; *Fox et al., 2022*; *Conte et al., 2018*).

For Muslim athletes, these congested game periods may cross the month of Ramadan, in which professional basketball players must continue their regular training and game routine, especially with no modified sports calendar during this period (*Brini et al., 2018*; *Kirkendall et al., 2008*). The RIF requires that adult Muslims complete only two daily meals (before sunrise and after sunset) (*Zerguini et al., 2007*). Additionally, the RIF is associated with significant changes in lifestyle rhythms (*Nobari et al., 2022*), body composition (*Trabelsi et al., 2022b*; *Brini et al., 2020*), sleep (*Trabelsi et al., 2023*; *Roky et al., 2004*), biochemical variables (*Brini et al., 2021a*; *Brini et al., 2021b*), and physical activity level, which could influence sport performance (*Brini et al., 2020*; *Nobari et al.,*

*2022*; *Brini et al., 2020*; *Roky et al., 2004*; *Brini et al., 2021a*; *Brini et al., 2021b*; *Chaouachi et al., 2008*; *Memari et al., 2011*). In this context, *Mujika, Chaouachi & Chamari (2010)* suggested Muslim athletes may reduce the training load during the RIF. Recently, *Brini et al. (2021a)* and *Brini et al. (2021b)* reported significantly greater decreases in body mass (4%), body fat percentage (9%), total energy intake (20%), carbohydrate intake (8%), protein intake (6%), sleep duration (29%), and sleep quality (22%) (using the Pittsburgh Sleep Quality Index (PSQI)) after 4 weeks of RIF in professional male basketball players. Moreover, these researchers reported that technical performances, body composition, sleep habits, and s-RPE were negatively affected. Based on what was reported above, it has been well demonstrated that the increased game frequency periods can influence the s-RPE and reduce the recovery time. Also, the RIF can be another factor influencing the state of fatigue and the recovery process in professional basketball players. Thus, it will be interesting to explore the effects of this increased game frequency period when it crosses the RIF to give more information to basketball coaches and physical trainers on how to deal with professional basketball players under these circumstances. To the best of our knowledge, no studies have yet investigated the specific impact of an increased game frequency period crossing the RIF on professional basketball players' performances.

Therefore, the aim of this study was to examine the changes in body composition, internal load, sleep habits, well-being indices, recovery state, and dietary intake with an increased game frequency (two *versus* one game per week) during the RIF in male professional basketball players. Based on previous studies (*Fox, Scanlan & Stanton, 2017*; *Fox et al., 2022*; *Conte et al., 2018*; *Brini, Ouerghi & Bouassida, 2020*), we hypothesized that the time coincidence of RIF and an increase in basketball games would negatively affect the sleep habit, well-being indices, recovery state, and dietary intake of professional male basketball players.

## MATERIELS AND METHODS

### Study design

A longitudinal study design was applied to assess the effects of increased game frequency during RIF on professional basketball players' body composition, dietary intake, sleep habits, well-being indices, and recovery state. Participants were divided into tow groups (an increased game frequency group [INCR, n = 14] and a normal game frequency group [NORM, n =14]). Overall, the study lasted five weeks and was conducted during the 2021-2022 basketball season, coinciding with the RIF (from April until May). The daily fasting length was 15–16 h (temperature: 25.3 ± 3.4 °C; relative humidity: 43.1 ± 9.8%; training session time: between 5:00 and 6:30 p.m.; game time: between 4:00 and 6:00 p.m.). During the RIF period, INCR trained four times per week and completed two games per week (Wednesday and Saturday), whereas NORM trained five times per week and completed one game per week (Saturday). Changes in weekly internal load (sum of all daily session RPE * session durations), dietary intake, body composition, well-being indices (sleep, fatigue, stress, delayed-onset-of-muscle-soreness (DOMS)), recovery state with the Total Quality Recovery (TQR) questionnaire, and sleep quality (PSQI) were investigated during the first and fourth weeks of the RIF for both groups (Fig. 1).

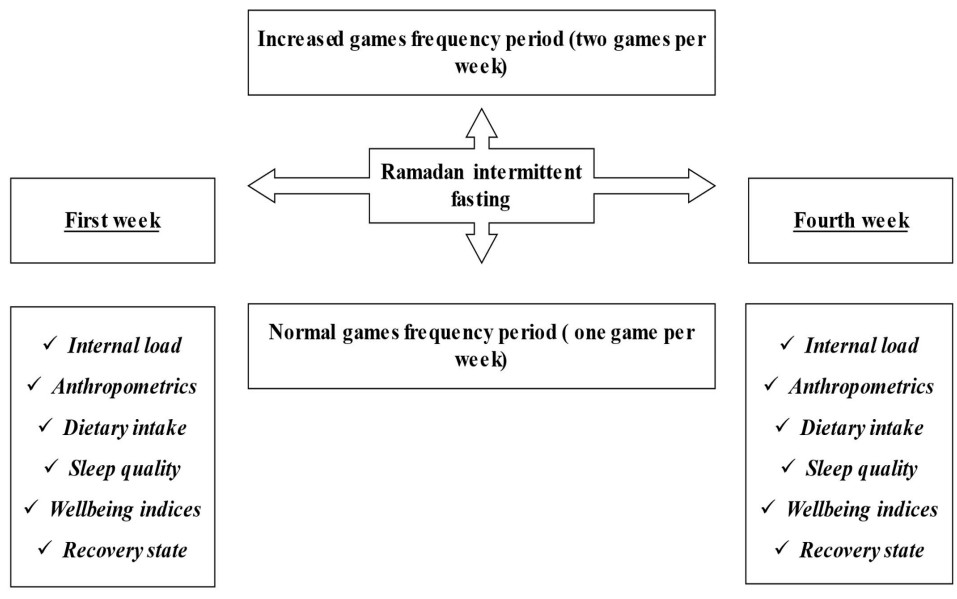

**Figure 1** Experimental protocol.

## Participants

The power of the simple size was calculated using G*Power *v.* 3.0 software (Heinrich Heine University Düsseldorf, Düsseldorf, Germany). A total of twenty-eight basketball players competing in the Tunisian league's first division volunteered to participate in this study (Table 1). For both groups, players had similar training experience (13.12 ± 1.21 years) and weekly practice load (≈8 h). The baseline physical fitness and technical skill levels were measured (*Delextrat & Kraiem, 2013*), and no significant differences between groups were found. Each team included three guards, three shooting guards, three small forwards, three forwards, and two centers. The inclusion criteria for study participation were: (1) participation in at least 90% of the training sessions during the RIF period; (2) participants were fasting all over the experimental period; (3) free of injuries the two months prior to RIF; and (4) not taking any supplements, medications, or other drugs. All participants were informed about the purposes, benefits, and risks associated with the study and provided written informed consent to participate. The investigation was approved by the local Clinical Research Ethics Committee of the Higher Institute of Sports and Physical Education of Kef, University of Jandouba, Kef, Tunisia (approval No. 4/2018), and the protocol was performed according to the Declaration of Helsinki.

## Procedures

Familiarization sessions were conducted for all participants before the beginning of the experimentation. Participants were instructed to wear the same footwear, and the training and testing sessions were conducted at the same time of day. All over the experimental period, training duration was ~90 min per session for both groups. Training sessions for INCR and NORM started with a warm-up, followed by basketball-specific drills
**Table 1 Anthropometric characteristics of the participating basketball players.**

| Groups | Age (years) | Height (m) | BM (kg) | BMI (kg.m$^{-2}$) |
|---|---|---|---|---|
| INCR ($n = 14$) | $26.2 \pm 2.1$ | $1.96 \pm 0.02$ | $88.6 \pm 6.1$ | $23.4 \pm 1.1$ |
| NORM ($n = 14$) | $25.8 \pm 2.1$ | $1.95 \pm 0.06$ | $88.7 \pm 6.8$ | $23.2 \pm 1.5$ |

Notes.
Data are reported as means and standard deviations.
INCR, increased game frequency group; NORM, normal game frequency group; BM, body mass; BMI, body mass index.

**Table 2 Weekly training program during the experimental period for the increased game frequency group (INCR) and the normal game frequency (NORM) group.**

| Days | Training program for INCR | Training program for the NORM |
|---|---|---|
| **Monday** | - Warm up, 15 min<br>- Specific basketball fundamental training, 15 min<br>- Moderate strength training (Upper/Lower), 20min<br>- Ball drill transition training ,15 min<br>- Technical/Tactical training, 20 min | - Warm up, 15 min<br>- Specific basketball fundamental training, 25 min<br>- Ball drill transition training ,15 min<br>- Technical/Tactical training, 30 min |
| **Tuesday** | - Warm up, 15 min<br>- Moderate intensity shooting exercises, 35 min<br>- Tactical training, 20 min<br>- Liveliness training, 10min<br>- Free throw shooting, 5 min | - Warm up, 15 min<br>- Moderate strength training (Upper/Lower), 25min<br>- Heigh intensity mid-range and<br>3 point shot exercises, 15 min<br>- Technical/Tactical training, 30 min |
| **Wednesday** | - Match of four 10 min quarters (FIBA) | - Warm up, 15 min<br>- Specific basketball fundamental training, 30 min<br>- Ball drill transition training, 20 min<br>- Three point shot exercises, 20 min |
| **Thursday** | - Warm up, 15 min<br>- Stretching (dynamic/static), 5min<br>- Free throw shooting, 10 min<br>- Moderate intensity shooting exercises, 30 min<br>- Technical/Tactical training, 25 min | - Warm up, 15 min<br>- Strength training (Lower body), 20min<br>- Moderate intensity mid-range<br>and 3 point shot exercises, 20 min<br>- Technical/Tactical training, 25 min |
| **Friday** | - Warm up, 15 min<br>- Low intensity 3 pts shooting exercises, 30 min<br>- Tactical training, 20min<br>- Liveliness training, 10min<br>- Free throw shooting, 10 min | - Warm up, 15 min<br>- Low intensity 3 pts shooting exercises, 30 min<br>- Tactical training, 15min<br>- Liveliness training, 10min<br>- Free throw shooting, 10 min |
| **Saturday** | - Match of four 10 min quarters (FIBA) | - Match of four 10 min quarters (FIBA) |
| **Sunday** | - recovery | - recovery |

(technical/tactical), specific basketball movements, basketball technique fundamentals, and basic defensive (Table 2).

## Assessment
### Internal load monitoring
The RPE was recorded for thirty minutes following each training session using Borg's 10-point Likert scale according to the methods described by *Foster et al. (2001)*. The s-RPE was calculated by multiplying the session duration (minutes) by the session RPE. The weekly load was then determined by summing the daily loads (s-RPE) for each athlete during each week.

Heart rate was continuously monitored throughout the training intervention using heart rate (HR) monitors (Polar Team Sport System; Polar-Electro OY, Kempele, Finland) and recorded at five-second intervals. The relative HR for each participant was obtained from the Yo-YoIR1. The following formula was used to calculate %HRmax: [%HRmax = (HRmean/HRmax) × 100]. Weekly HRavg was calculated by the average value for the entire week for each period.

## Anthropometrics

Height was measured in cm with a precision of 0.1 cm using a portable stadiometer (Seca, Maresten, UK). Participants were weighed wearing light clothes and without shoes using an electronic balance (Soehnle Pharo 200 Analytic; Soehnle Industrial Solutions GmbH, Backnang, Germany) with a precision of 0.1 kg, and the body mass index (BMI) was calculated by dividing the weight (kg) by the square of the height (m). The four-site skinfolds method (biceps, triceps, sub-scapular, and suprailliac skin folds) was used to determine the body fat percentage. A caliper (Harpenden caliper) was utilized during this technique. Body fat percent-age was then calculated: BF% = ((4.95/body density) −4.5) × 100 (body density = 1.1631−0.0632.log of the sum of four skin folds) (*Sisic et al., 2015*). Anthropometric measurements were taken according to the International Society for the Advancement of Kinanthropometry (ISAK) recommendations. The same qualified technician took all these measures.

## Dietary intake

Participants noted all meals consumed throughout the experimental period, noting and recording the amounts and types of food and fluid consumed. Additionally, they were interviewed, and a nutritionist analyzed the data by means of the software program Bilnut (Nutrisoft Bilnut: Food Survey Program version 2.01) and the food composition tables of the Tunisian National Institute of Statistics (*Hammouda et al., 2013*).

## The Pittsburgh sleep quality index

The validated Arabic version of the Pittsburgh Sleep Quality Index (*Suleiman et al., 2009*) was used to assess subjective sleep quality during the RIF (*Buysse et al., 1989*). It is comprised of 19 questions covering seven components of sleep: duration, quality, latency, efficiency, disturbances, daytime dysfunction, and the use of sleeping medications. The total score ranges from 0 to 21, where "0–4" designates good sleep and "5–21" indicates poor sleep in all sleep areas.

## Psychometric markers

Fifteen minutes before the warm-up, each player was asked to complete ratings of well-being indices (quality of sleep, fatigue, stress, and delayed onset of muscle soreness [DOMS]), considering the timeline from the last (training or match) session until the moment of the new session. Players rated each index using a scale from 1 to 7 points, where one indicated "very, very low" (fatigue, stress, and DOMS) or "good" (quality of sleep) and 7 indicated "very, very high" (fatigue, stress, and DOMS) or "bad" (quality of sleep) (*Sampaio, Abrantes & Leite, 2009*). The sum of these four scores was used to calculate

the Hooper Index (HI) (*Ouergui et al., 2020a*; *Ouergui et al., 2020b*). A higher HI score indicates a more negative state of well-being. After each player completed the well-being indices, the recovery state was evaluated using the Total Quality Recovery (TQR) scale (*Selmi et al., 2018*). The TQR scale ranges from 6 to 20, where 6 indicates "very, very poor recovery" and 20 indicates "very, very good recovery". Previous studies have used this scale as an indicator of athletes' perceived recovery (*Selmi et al., 2018*).

## Statistical analyses

The statistical analyses were computed using SPSS for Windows, version 20.0 (SPSS Inc., Chicago, IL, USA). The cutoff esteem was set to be $p \leq 0.05$ for all measures to represent statistical significance. The normality was tested and confirmed using the Shapiro–Wilk test, and the data were presented as means and standard deviations (SD). Baseline values were compared using t-tests for independent samples. The variation of all measures was compared using a 2 (groups: INCR, NORM) × 2 (time: week 1, week 4) repeated measures ANOVA. The Bonferroni-corrected post-hoc tests were performed if significant interactions were found. Additionally, effect sizes (ES) were determined from ANOVA output by converting partial eta-squared to *Cohen*'s d *1973*. Following *Hopkins et al. (2009)*, ES were considered trivial (<0.2), small (0.2–0.6), moderate (0.6–1.2), large (1.2–2.0), and very large (2.0–4.0). In order to examine the relationship between weekly internal load, sleep habits, dietary intakes, recovery state, and well-being indices at the end of the experimental period, Pearson's linear correlation coefficient was used. The magnitude of the correlation was expressed as trivial: $r < 0.1$; low: 0.1–0.3; moderate: 0.3–0.5; large: 0.5–0.7; very large: 0.7–0.9; nearly perfect >0.9; and perfect: 1.

## RESULTS

All through the experimentation, we didn't notice several injuries. There were significant differences between groups concerning adherence (97.8% for INCR and 97.6% for NORM) and playing time per game (21.3 ± 1.2 min for INCR and 20.8 ± 1.9 min for NORM).

Changes in body composition and estimated daily dietary intake during the experimental period are presented in Table 3. Significant group × time interaction was observed for BM ($p = 0.006$, ES = 0.46, small), and BF% ($p = 0.025$, ES = 0.33, small), with INCR having a greater decrease in all these values. Bonferroni-corrected post hoc tests revealed a significant decrease in BM, FM, and BF% at the end of the experimental period for both groups (INCR: 3.63%, $p < 0.001$, ES = 0.24, small; NORM: 2.39%, $p < 0.001$, ES = 0.31, small); and BF% (INCR: 7.99%, $p < 0.001$, ES = 0.21, small; NORM: 2.74%, $p < 0.05$, ES = 0.18, trivial; respectively).

A significant group × time interaction was observed for a greater decrease in carbohydrate intake in INCR compared to NORM ( $p = 0.010$, ES = 0.44, small). Bonferroni-corrected post hoc tests revealed a significant decrease for the INCR (8.32%, $p < 0.001$, ES = 3.75, large).

Table 4 shows changes in quality of sleep, well-being indices, recovery state, and internal load after 4 weeks of RIF. There was a significant group x time interaction for RPE ($p = 0.025$, ES = 0.33, small) with INCR having a larger increase after 4 weeks of RIF.

**Table 3  Body composition and estimated daily dietary intake recorded during the first and the fourth weeks of Ramadan intermittent fasting.**

| Variables | | First week | | Fourth week | | Pvalue (ES) | | |
|---|---|---|---|---|---|---|---|---|
| | | INCR | NORM | INCR | NORM | Time | Group | Group × Time |
| Body mass (kg) | | 88.6 ± 6.1 | 88.7 ± 6.8 | 85.4 ± 5.9 | 86.6 ± 6.6 | <0.001(0.92) | 0.79(0.01) | 0.01(0.46) |
| BMI (kg/m² ) | | 23.4 ± 1.1 | 23.2 ± 1.5 | 23.3 ± 1.1 | 23.3 ± 1.5 | <0.001(0.90) | 0.72(0.01) | 0.07(0.24) |
| Body fat (%) | | 12.1 ± 2.3 | 12.3 ± 2.5 | 11.1 ± 2.0 | 12.0 ± 2.5 | <0.001(0.65) | 0.28(0.09) | 0.03(0.33) |
| FM (kg) | | 10.68 ± 2.05 | 10.96 ± 2.54 | 9.43 ± 1.69 | 10.40 ± 2.44 | <0.001(0.65) | 0.23(0.10) | 0.016(0.37) |
| FFM (kg) | | 77.97 ± 6.05 | 77.76 ± 6.10 | 75.98 ± 5.92 | 76.17 ± 5.80 | 0.993(0.00) | <0.001(0.79) | 0.249(0.10) |
| Total energy intake (kcal) | | 2766.4 ± 173.0 | 2811.4 ± 92.5 | 2217.9 ± 183.7 | 2369.3 ± 175.4 | <0.001(0.90) | 0.02(0.34) | 0.25(0.10) |
| Carbohydrate intake | (g) | 310.7 ± 19.8 | 309.3 ± 16.9 | 284.3 ± 16.0 | 297.1 ± 14.9 | 0.002(0.54) | 0.30(0.08) | 0.01(0.41) |
| | (%) | 45.1 ± 3.6 | 44.0 ± 2.6 | 51.6 ± 5.6 | 50.4 ± 3.9 | <0.001(0.70) | 0.31(0.08) | 0.91(0.001) |
| Protein intake | (g) | 88.2 ± 5.0 | 86.5 ± 3.9 | 83.4 ± 4.7 | 83.0 ± 3.8 | <0.001(0.83) | 0.51(0.04) | 0.20(0.12) |
| | (%) | 12.8 ± 1.1 | 12.3 ± 0.6 | 15.1 ± 1.3 | 14.1 ± 1.4 | <0.001(0.80) | 0.05(0.26) | 0.36(0.06) |
| Fat intake | (g) | 122.9 ± 11.4 | 123.2 ± 9.3 | 95.1 ± 7.4 | 103.1 ± 12.7 | <0.001(0.81) | 0.11(0.18) | 0.07(0.10) |
| | (%) | 40.1 ± 4.3 | 39.5 ± 2.8 | 38.9 ± 4.3 | 39.1 ± 3.7 | 0.55(0.03) | 0.83(0.004) | 0.60(0.02) |

**Notes.**
Data are reported as means and standard deviations.
INCR, increased game frequency group; NORM, normal game frequency group; BMI, body mass index; FM, fat mass; FFM, fat free mass.

Bonferroni-corrected post hoc tests revealed a significant increase for both groups in RPE (INCR: −22.63%, $p < 0.001$, ES = 0.16, trivial; NORM: −12.52%, $p < 0.001$, ES = 0.15, trivial; respectively). There was also a significant group × time interaction for HR average ($p = 0.002$, ES = 0.55, small), with INCR having a greater decrease. Bonferroni-corrected post hoc tests revealed a significant decrease in HR average for both groups (INCR: 1.54%, $p < 0.001$, ES = 0.25, small; NORM: 0.79%, $p < 0.001$, ES = 0.16, trivial; respectively). There were significant group × time interactions for sleep duration, TQR, and DOMS ( $p = 0.009$, ES = 0.41, small; $p = 0.039$, ES = 0.29, small; $p = 0.045$, ES = 0.28, small; respectively), with INCR having the greater decrease in sleep duration and TQR and greater increases in DOMS. Bonferroni-corrected post hoc tests revealed a significant decrease in TQR and sleep duration and a significant increase in DOMS only for INCR (26.93%, $p < 0.001$, ES =0.48, small; 33.83%, $p < 0.001$, ES = 0.40, small; 161.17%, $p < 0.001$, ES = 0.32, small; respectively).

Correlation analysis showed that BF% was negatively correlated with HR average ($p = 0.024$, $r = −0.42$, large magnitude), DOMS was positively correlated with RPE ($p = 0.024$, $r = 0.42$, large magnitude), and sleep duration was positively correlated with percentage of changes in HR ($p = 0.016$, $r = 0.45$, large magnitude).

# DISCUSSION

The current investigation aimed to assess the effects of an increased game frequency periods during RIF on professional basketball players' body composition, sleep habits, well-being indices, recovery state, and dietary intake. The main results showed a greater decrease in body mass, body fat %, fat mass, carbohydrate intake, sleep duration, average session HR, and recovery in INCR *versus* NORM. INCR also had greater increases in session RPE

**Table 4** Measurement of the subjective quality of sleep, well-being indices, recovery state and internal load recorded during the first and the fourth weeks of Ramadan intermittent fasting.

| Variables | First week | | Fourth week | | P value (ES) | | |
| --- | --- | --- | --- | --- | --- | --- | --- |
| | INCR | NORM | INCR | NORM | Time | Group | Group × Time |
| **Quality of sleep** | | | | | | | |
| Sleep latency (min) | 14.64 ± 1.01 | 14.78 ± 0.89 | 14.71 ± 1.99 | 14.93 ± 1.14 | 0.43(0.05) | 0.58(0.02) | 0.79(0.01) |
| Sleep efficiency (%) | 94.21 ± 0.69 | 94.79 ± 0.89 | 94.43 ± 1.45 | 94.78 ± 1.53 | 0.66(0.02) | 0.13(0.16) | 0.79(0.01) |
| Sleep duration (h) | 9.78 ± 153 | 9.64 ± 1.50 | 7.00 ± 1.11 | 8.07 ± 0.99 | <0.001(0.67) | 0.12(0.18) | 0.01(0.41) |
| Total score of PSQI | 4.64 ± 1.01 | 4.72 ± 0.82 | 4.28 ± 1.06 | 4.71 ± 0.83 | 0.46(0.05) | 0.35(0.0.07) | 0.47(0.04) |
| **Well-being indices** | | | | | | | |
| Sleep (1–7) | 3.92 ± 0.46 | 4.14 ± 0.69 | 6.35 ± 0.36 | 6.24 ± 0.38 | <0.001(0.95) | 0.72(0.01) | 0.14(0.16) |
| Fatigue (1–7) | 3.86 ± 0.44 | 4.27 ± 0.64 | 5.65 ± 0.58 | 6.04 ± 0.78 | <0.001(0.91) | 0.03(0.31) | 0.95(0.000) |
| Stress (1–7) | 3.61 ± 1.29 | 4 ± 1.03 | 5.88 ± 0.51 | 5.72 ± 0.61 | <0.001(0.84) | 0.68(0.01) | 0.24(0.10) |
| DOMS (1–7) | 2.20 ± 0.60 | 2.53 ± 0.59 | 5.25 ± 0.83 | 4.77 ± 0.53 | <0.001(0.92) | 0.49(0.04) | 0.05(0.28) |
| HI | 12.94 ± 1.69 | 14.21 ± 2.49 | 22.94 ± 1.53 | 22.73 ± 1.73 | <0.001(0.95) | 0.30(0.08) | 0.18(0.13) |
| **Recovery state** | | | | | | | |
| TQR (out of 28) | 13.66 ± 1.12 | 12.85 ± 1.44 | 8.96 ± 0.66 | 9.34 ± 1.06 | <0.001(0.91) | 0.38(0.06) | 0.04(0.29) |
| **Internal load** | | | | | | | |
| RPE (0–10) | 5.91 ± 0.40 | 5.93 ± 0.53 | 7.21 ± 0.33 | 6.63 ± 0.33 | <0.001(0.90) | 0.02(0.34) | 0.03(0.33) |
| W load | 2359.29 ± 112.62 | 2.397.85 ± 171.51 | 2648.56 ± 98.05 | 2565 ± 84.65 | <0.001(0.76) | 0.47(0.04) | 0.10(0.19) |
| HR average | 167.45 ± 2.30 | 166.22 ± 2.56 | 164.86 ± 2.11 | 164.90 ± 2.60 | <0.001(0.95) | 0.42(0.05) | 0.002(0.52) |
| %HR max | 83.79 ± 1.46 | 83.06 ± 1.47 | 82.50 ± 1.28 | 82.39 ± 1.48 | <0.001(0.94) | 0.40(0.05) | 0.002(0.52) |

**Notes.**
Data are reported as means and standard deviations.

INCR, increased game frequency group; NORM, normal game frequency group; HR, heart rate; RPE, The rating of perceived exertion; PSQI, sleep quality index; DOMS, delayed onset of muscle soreness; TQR, Total Quality Recovery; W load, weekly load; HI, Hooper Index (sum of scores for sleep, fatigue, stress and DOMS).

and DOMS at the end of the experimental period compared to NORM. In general, the combination of RIF and increased basketball game frequency periods negatively affected sleep, well-being, recovery state, and dietary intake in professional basketball male players.

## Weekly internal load

The weekly load values obtained during the current study were similar to those reported in prior studies in basketball fields (*Fox, Scanlan & Stanton, 2017*; *Fox, Scanlan & Stanton, 2017*), which were conducted during the regular competitive season period (2400 AU). However, our values were higher than observed in basketball players before and after RIF in a study by *Brini et al. (2021a)* and *Brini et al. (2021b)*. Concerning the s-RPE variability, our findings showed a significant increase in s-RPE scores at the end of the RIF for both groups, with a significantly higher increase in favor of INCR. The findings of the present investigation may be explained by the increased muscle fatigue, stress, and muscle damage accumulated at the end of RIF following the increased game periods. This increased muscle damage was due to the greater intensity of games combined with lower carbohydrate ingestion and thus possibly lower muscle glycogen levels. Our results concord with previous research that reported decreased muscle power and higher RPE during RIF (*Chtourou et al., 2011*; *Trabelsi et al., 2022b*). In the same context, *Leiper et al.*

*(2008)* reported that decreases in physical function can increase perceived exertion and lead to an earlier onset of fatigue, which may increase the risk of injury.

Concerning HR average and %HRmax, the present investigation revealed a significant decrease at the end of Ramadan for both groups compared with the first week. This decrease in the HR may be justified by the accumulated muscle fatigue observed at the end of the RIF. The lower recovery state associated with a higher DOMS index resulted in more significant fatigue in the players and a lack of high-performance effort during training and games. Internal (hydration status, emotional state), environmental (temperature, humidity), technical, and activity-specific factors may influence HR responses. For example, HR responses may vary due to natural variation since HR data was collected across multiple training sessions or games spanning several weeks. In the same context, *Al Suwaidi et al. (2006)* explained this decrease by the interaction between the fast state, catecholamine inhibition, and reduced venous return, which may decrease the sympathetic tone and lead to a decrease in blood pressure, HR, and cardiac output.

### Sleep quality

Several investigations showed the importance of sleep as a crucial component of successful training, competition, and recovery (*Samuels, 2008*; *Trabelsi et al., 2022a*). In this sense and following significant diurnal changes, Muslim athletes tend to prepare for RIF through several habits' changes (*i.e.,* sleeping and eating time and frequency) (*Waterhouse et al., 2009*). As a result, they likely suffer from sleep loss or fragmentation, which may reduce athletic performance (*Waterhouse et al., 2009*). The result of the present study revealed a negative effect of the RIF only on sleep duration, which supports the findings of *Brini et al. (2021a)* and *Brini et al. (2021b)* in a prior study conducted in professional basketball players. The decrease in sleep duration was greater in the INCR group. In the same context, *Roky et al. (2004)* reported that sleep loss accumulated during RIF may disrupt the sleep-wake cycle, leading to more fatigue and reducing mental and physical performance. This information must be considered by practitioners in order to adapt their recovery strategies during RIF.

### Body composition and dietary intakes

Our results revealed a decrease in BM, FM, and BF% for both groups recorded at the end of the RIF. Our findings are in line with previous studies in athletes and basketball players, more specifically (*Trabelsi et al., 2023*; *Brini et al., 2020*; *Memari et al., 2011*). In general, these variations are due to changes in lifestyle during the RIF in variables such as sleep hours, physical activities, food consumption, frequency of meals, and dietary patterns for different reasons, especially for athletes who try to maintain their physical performance during this extreme and exhaustive period. In addition, the significant body mass reduction noticed at the end of the RIF in the current study may also be attributed to the decrease in fluid intake (*Baklouti et al., 2017*; *Beltaifa et al., 2004*), the decrease in glycogen-bound water stores (*Chtourou et al., 2011*), and the reduction in total energy and macronutrient intake. Moreover, this observed decline in body composition is attributed to a decrease in fluid intake and hypohydration with little body fat loss (*Brini et al., 2020*; *Chaouachi et al.,*

*2008*; *Memari et al., 2011*). Future studies exploring the game-frequency-dependent effects of RIF on body composition should strictly measure the hydration status of participants to better account for the potential confounding influence of this factor on study results.

## Markers of well being

The most interesting findings about the wellbeing markers were the reported increase in DOMS and decrease in TQR values, which were more negatively affected in the fourth week compared with the beginning of RIF for INCR. Our results were similar to previous studies in team sports and basketball (*Brini et al., 2021a*; *Brini et al., 2021b*). Those studies reported that changes in the daily habits of players during RIF, such as rising earlier and eating a meal before sunrise, may be affected by partial sleep deprivation, which will affect the higher cognitive centers of the central nervous system, which negatively affect mental activity, which has been posited as one of the main reasons for performance declines (*Chtourou et al., 2011*; *Fashi et al., 2021*; *Smith et al., 2016*). In the same context, *Davis (2000)* reported that fatigue might reduce muscle glycogen depletion and/or alter neurotransmitter activity, which could, in turn, negatively influence cognition and motor skill performance. The lower carbohydrate intake with INCR could also have negatively affected muscle and liver glycogen stores and, thus, cognitive function.

Finally, our investigation has some limitations. Firstly, some parameters, such as the time-motion analysis and the shooting accuracy, were not measured to explore the impacts of fatigue accumulated during the increased game period crossing the RIF. Secondly, the nature and number of participants used can also affect the results obtained since it is very possible that with the exploration of a larger population that evolves at other levels, we will have other results. Thirdly, the addition of other groups (groups without doing Ramadan, control groups without games and training, *etc.*). Finally, future studies may produce different results by investigating the RIF under other climatic circumstances, in both sexes, and with different sports.

## CONCLUSIONS

Our results showed that body composition, sleep duration, carbohydrate intake, DOMS, and TQR were significantly affected at the end of RIF in association with the changes in weekly internal load, specifically by adding one more game per week. Increasing carbohydrate intake, reducing training session duration, or adding other recovery methods could improve sleep and reduce the increased muscle soreness experienced with increased game frequency crossing RIF. The present study's findings should be considered by researchers and practitioners when players are experiencing congested basketball games during RIF.

## ACKNOWLEDGEMENTS

The authors would like to acknowledge, with considerable gratitude, all those who volunteered to participate in this study.

### Funding

Javier Raya-González was supported by a Ramón y Cajal postdoctoral fellowship (RYC2021-031072-I) given by the Spanish Ministry of Science and Innovation, the State Research Agency (AEI) and the European Union (Next Generation EU/PRTR). The funders had no role in study design, data collection and analysis, decision to publish, or preparation of the manuscript.

### Grant Disclosures

The following grant information was disclosed by the authors:
Ramón y Cajal postdoctoral fellowship: RYC2021-031072-I.
Spanish Ministry of Science and Innovation.
State Research Agency (AEI).
European Union (Next Generation EU/PRTR).

### Competing Interests

Luca Paolo Ardigò is an Academic Editor for PeerJ.

### Author Contributions

- Seifeddine Brini conceived and designed the experiments, performed the experiments, analyzed the data, prepared figures and/or tables, authored or reviewed drafts of the article, and approved the final draft.
- Luca Paolo Ardigò conceived and designed the experiments, analyzed the data, prepared figures and/or tables, authored or reviewed drafts of the article, and approved the final draft.
- Filipe Manuel Clemente conceived and designed the experiments, analyzed the data, prepared figures and/or tables, authored or reviewed drafts of the article, and approved the final draft.
- Javier Raya-González analyzed the data, prepared figures and/or tables, authored or reviewed drafts of the article, and approved the final draft.
- Jennifer A. Kurtz analyzed the data, authored or reviewed drafts of the article, and approved the final draft.
- Gretchen A. Casazza conceived and designed the experiments, analyzed the data, authored or reviewed drafts of the article, and approved the final draft.
- Carlo Castagna conceived and designed the experiments, analyzed the data, authored or reviewed drafts of the article, and approved the final draft.
- Anissa Bouassida conceived and designed the experiments, performed the experiments, prepared figures and/or tables, and approved the final draft.
- Hadi Nobari conceived and designed the experiments, analyzed the data, prepared figures and/or tables, authored or reviewed drafts of the article, and approved the final draft.

## Ethics

The following information was supplied relating to ethical approvals (*i.e.*, approving body and any reference numbers):

The investigation was approved by the local Clinical Research Ethics Committee of the Higher Institute of Sports and Physical Education of Kef, University of Jandouba, kef, Tunisia (approval No. 4/2018) and the protocol was performed according to the Declaration of Helsinki.

## Data Availability

The raw measurements are available in the Supplemental File.

## Supplemental Information

Supplemental information for this article can be found online at http://dx.doi.org/10.7717/peerj.16507#supplemental-information.

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
