# Peer review of "Increased game frequency period crossing Ramadan intermittent fasting decreases fat mass, sleep duration, and recovery in male professional basketball players"

_PeerJ, doi:10.7717/peerj.16507_

## Round 0.1 · original submission · Major Revisions

Dear Authors,
The reviewers and I have completed our evaluation of your manuscript and recommend a major revision before re-submission.

Please review the comments and resubmit your revised manuscript.

Reviewer 1 has suggested that you cite specific references. You are welcome to add it/them if you believe they are relevant. However, you are not required to include these citations, and if you do not include them, this will not influence my decision.

**Language Note:** The review process has identified that the English language must be improved. PeerJ can provide language editing services - please contact us at copyediting@peerj.com for pricing (be sure to provide your manuscript number and title). Alternatively, you should make your own arrangements to improve the language quality and provide details in your response letter. – PeerJ Staff

Reviewer 1 ·

Basic reporting

The aim of the present investigation was to assess the impacts of increased game frequency, as a measure of increased physiological load, on body composition, sleep habits, indices of well-being, recovery state and dietary intake in professional male basketball players during Ramadan intermittent fasting (RIF).
The authors concluded that Basketball players are significantly affected by increased game frequency (physiological load) during Ramadan with larger decreases in sleep duration, carbohydrate intake, and recovery state and increased DOMS; all of which could affect exercise performance and health.
The manuscript is well written and present interesting findings. However, some modifications are required:


The rational of the study should be explained using recent systematic reviews or meta analysis and/or recent studies. Considering the main topic, I suggest:
Trabelsi, K., Ammar, A., Glenn, J. M., Boukhris, O., Khacharem, A., Bouaziz, B., ... & Chtourou, H. (2022). Does observance of Ramadan affect sleep in athletes and physically active individuals? A systematic review and meta‐analysis. Journal of Sleep Research, 31(3), e13503.
Trabelsi, K., Ammar, A., Boukhris, O., Glenn, J. M., Clark, C. C., Stannard, S. R., ... & Chtourou, H. (2023). Dietary intake and body composition during Ramadan in athletes: a systematic review and meta-analysis with meta-regression. Journal of the American Nutrition Association, 42(1), 101-122.
Trabelsi, K., Ammar, A., Boujelbane, M. A., Khacharem, A., Elghoul, Y., Boukhris, O., ... & Terry, P. C. (2022). Ramadan observance is associated with higher fatigue and lower vigor in athletes: a systematic review and meta-analysis with meta-regression. International Review of Sport and Exercise Psychology, 1-28.
Trabelsi, K., Bragazzi, N., Zlitni, S., Khacharem, A., Boukhris, O., El-Abed, K., ... & Chtourou, H. (2020). Observing Ramadan and sleep-wake patterns in athletes: a systematic review, meta-analysis and meta-regression. British Journal of Sports Medicine, 54(11), 674-680.
Boukhris, O., Trabelsi, K., Shephard, R. J., Hsouna, H., Abdessalem, R., Chtourou, L., ... & Chtourou, H. (2019). Sleep patterns, alertness, dietary intake, muscle soreness, fatigue, and mental stress recorded before, during and after Ramadan observance. Sports, 7(5), 118.
Chtourou, H., Trabelsi, K., Boukhris, O., Ammar, A., Shephard, R. J., & Bragazzi, N. L. (2019). Effects of Ramadan fasting on physical performances in soccer players: A systematic review effets du jeûne de Ramadan sur les performances physiques des footballeurs: Revue systématique. Tun Med, 97, 1114-1131.
Aloui, A., Baklouti, H., Souissi, N., & Chtourou, H. (2019). Effets du jeûne du Ramadan sur la composition corporelle des sportifs: revue systématique.
As these studies are directly related to the topic, I suggest that they are usefull also for the discussion.

A short section about Ramadan should be added and containing: the year, the fasting duration, the environmental condition, the moments or data collection, …
A short part in the discussion presenting all limitations is important (e.g., the absence of the control group, …).

Experimental design

Well written and prepared.
Only, a short section about Ramadan should be added and containing: the year, the fasting duration, the environmental condition, the moments or data collection, …

Validity of the findings

The data are important and have direct applications

Reviewer 2 ·

Basic reporting

I would thanks the authors for the effort and for the best quality of the manuscript. However, there is some mistakes to correct them.
General comments:
- The authors need to work with a fluent English speaker/writer to correct grammatical and punctuation errors throughout the manuscript.

- Title of paper need to be reformulated

Abstract
- The aim of the study is not clear
- Results: should be more detailed
- Conclusion need to be reorganized
Introduction
- L75-L77 : More details and information about basketball
- L90-L93: Reformulate this paragraph
- The problematic is not clear

Experimental design

Methods
- L75: change “Heigh” to “Higher”
- More information about the heart rate monitors used.
- Has the food control been done well? It can affect results or not ?
- Does lack of sleep affect performance? how did you deal with this lack?
Results
- Table 3 and 4 are not very clear. They must be simplified to become clear to the reader.

Validity of the findings

Discussion
- L279-L281: Add reference
- L284-L290: This justification is not clear and need to be more detailed
- Discuss limitations of the study, taking into account sources of potential bias or imprecision. Discuss both direction and magnitude of any potential bias
- Discuss the generalisability (external validity) of the study results
- Discuss the practical implications and future research

·

Basic reporting

no comments

Experimental design

no comments

Validity of the findings

no comments

Additional comments

no comments

---

## Round 0.2 · accepted · Accept

Your manuscript has been accepted for publication. Congratulations!

Reviewer 2 ·

Basic reporting

Mistakes are well corrected. The manuscript now is clear and i think is suitable for publication

Experimental design

Mistakes are well corrected. The manuscript now is clear and i think is suitable for publication

Validity of the findings

Mistakes are well corrected. The manuscript now is clear and i think is suitable for publication

Additional comments

Mistakes are well corrected. The manuscript now is clear and i think is suitable for publication